# Environment- and epigenome-wide association study of obesity in 'Children of 1997' birth cohort

Jie Zhao[1]*[†], Bohan Fan[1][†], Jian Huang[2], Benjamin John Cowling[1], Shiu Lun Ryan Au Yeung[1], Andrea Baccarelli[3], Gabriel M Leung[1], C Mary Schooling[1,4]

[1]School of Public Health, Li Ka Shing Faculty of Medicine, The University of Hong Kong, Hong Kong, Hong Kong; [2]Singapore Institute for Clinical Sciences (SICS), Agency for Science, Technology and Research (A*STAR), Singapore, Singapore; [3]Mailman School of Public Health, Columbia University, New York, United States; [4]City University of New York, School of Public Health and Health policy, New York, United States

## Abstract

**Background:** Increasing childhood obesity is a global issue requiring potentially local solutions to ensure it does not continue into adulthood. We systematically identified potentially modifiable targets of obesity at the onset and end of puberty in Hong Kong, the most economically developed major Chinese city.

**Methods:** We conducted an environment-wide association study (EWAS) and an epigenome-wide association study of obesity to systematically assess associations with body mass index (BMI) and waist–hip ratio (WHR) in Hong Kong's population-representative 'Children of 1997' birth cohort. Univariable linear regression was used to select exposures related to obesity at ~11.5 years (BMI and obesity risk $n \leq 7119$, WHR $n = 5691$) and ~17.6 years ($n = 3618$) at Bonferroni-corrected significance, and multivariable regression to adjust for potential confounders followed by replicated multivariable regression ($n = 308$) and CpG by CpG analysis ($n = 286$) at ~23 years. Findings were compared with evidence from published randomized controlled trials (RCTs) and Mendelian randomization (MR) studies.

**Results:** At ~11.5 and ~17.6 years the EWAS identified 14 and 37 exposures associated with BMI, as well as 7 and 12 associated with WHR, respectively. Most exposures had directionally consistent associations at ~23 years. Maternal second-hand smoking, maternal weight, and birth weight were consistently associated with obesity. Diet (including dairy intake and artificially sweetened beverages), physical activity, snoring, binge eating, and earlier puberty were positively associated with BMI at ~17.6 years, while eating before sleep was inversely associated with BMI at ~17.6 years. Findings for birth weight, dairy intake, and binge eating are consistent with available evidence from RCTs or MR studies. We found 17 CpGs related to BMI and 17 to WHR.

**Conclusions:** These novel insights into potentially modifiable factors associated with obesity at the outset and the end of puberty could, if causal, inform future interventions to improve population health in Hong Kong and similar Chinese settings.

**Funding:** This study including the follow-up survey and epigenetics testing was supported by the Health and Medical Research Fund Research Fellowship, Food and Health Bureau, Hong Kong SAR Government (#04180097). The DNA extraction of the samples used for epigenetic testing was supported by CFS-HKU1.

*For correspondence:
janezhao@hku.hk

†These authors contributed equally to this work

## Editor's evaluation

This study presents a valuable finding on the association of environmental and epigenomic factors with obesity in the adolescent population, using a specialized cohort. The evidence supporting the claims of the authors is solid, although this study is inevitably open to residual confounding because of the limitation of observational study. The work will be of interest to both clinicians and researchers working on obesity and related metabolic disorders.

## Introduction

With improving living standards and socioeconomic development, non-communicable chronic diseases pose a heavy burden on society in both developed and developing countries (*Lozano et al., 2012*). Obesity is a well-established risk factor for multiple chronic diseases, including cardiovascular disease, diabetes, and cancer (*Gallagher and LeRoith, 2015*). According to the World Health Organization (WHO), obesity is defined as 'abnormal or excessive fat accumulation that presents a risk to health' (*World Health Organization, 2000*). Body mass index (BMI) is the most appropriate measure for overweight and obesity because the cut-offs account for age, sex, and ethnicity (*World Health Organization, 2000*). Obesity has increased substantially in many settings, including in Hong Kong. Obesity is complicated with multifactorial risk factors, such as socioeconomic position (SEP), mood disturbance, and genetic factors (*Cardel et al., 2020*), so there is a need to evaluate their roles in obesity comprehensively and systematically (*Manrai et al., 2017*). Moreover, given most studies on targets of obesity are conducted in a western setting (*Cardel et al., 2020*), a comprehensive assessment of modifiable factors of early life obesity in a non-western setting, with a different social structure, provides a valuable opportunity to identify novel exposures.

Environment-wide association studies (EWAS) enable us to assess a variety of exposures across the human environmental exposome in a high-throughput manner (*Hall et al., 2013*), similar to genome-wide association studies for genetic associations. Previous EWAS have been performed on outcomes, such as type 2 diabetes (*Patel et al., 2010*), cardiovascular disease (*Zhuang et al., 2018*), and childhood obesity in western settings (*Vrijheid et al., 2020*; *Uche et al., 2020*) but not in a Chinese setting. In the previous EWAS of childhood obesity in the US (6–17 years old), UK, and Europe (6–11 years old), second-hand smoking was related to higher childhood BMI, whilst some other exposures, such as vitamins, were not consistently associated with obesity in these settings (*Vrijheid et al., 2020*; *Uche et al., 2020*). Observational studies are open to confounding by SEP, thus assessing associations in a different social context can triangulate the evidence concerning early life obesity.

In addition to the environmental factors, it is increasingly realized that epigenetic factors, which are also modifiable, may play an important role in obesity (*Huang et al., 2018a*). DNA methylation, which refers to the addition of a methyl group to the 5′ position of a cytosine residue of the DNA, is the most frequently examined epigenetic modification (*Fall et al., 2017*). DNA methylation may modulate gene expression and thereby influence susceptibility to obesity or obesity-related chronic disease (*Fall et al., 2017*). Epigenome-wide association study provides an approach to identify the related epigenetic loci in a comprehensive way. For example, DNA methylation at cg06500161 was previously identified as related to obesity in the US (*Huang et al., 2018a*). Unlike genetic variants, DNA methylation is modifiable and may change in response to environmental factors or disease and therefore might be open to confounding by these factors (*Fall et al., 2017*; *Fraga et al., 2005*). As such, findings from western settings may not be generalizable to Chinese populations.

In this situation, Hong Kong, a non-western developed setting, can provide unique insights into health determinants. Most Chinese people in Hong Kong are first-, second-, or third-generation migrants from the neighbouring province of Guangdong in southern China. Dietary habits of people in Hong Kong are similar to those in southern China, although also influenced by western culture (*Leung et al., 2003*). Lifestyle in Hong Kong also differs from more commonly studied western populations on some important attributes, for example, active smoking among Chinese mothers was rare while maternal exposure to second-hand smoking during pregnancy was common before the smoking ban in public and workplaces was implemented in 2007 (*Lee, 2016*). Moreover, most current theories concerning the aetiology of and disparities in chronic diseases originate from observations in long-term developed populations of European descent. However, in Hong Kong the economic transition from pre- to post-industrial living conditions has occurred within one lifetime of the older people

(*Leung et al., 2017*), whereas children today in Hong Kong represent the first generation of Chinese to grow up in a post-industrial Chinese setting, which is unrivalled anywhere in the world (*Schooling et al., 2012*). As such, a study in young Chinese people in Hong Kong, a different setting provides 'a sentinel for populations currently experiencing very rapid economic development' (*Schooling et al., 2016*), which may help identify whether these associations reflect SEP within a specific context or are biologically based as well as having the potential to identify any attributes relevant to the majority of the global population but not necessarily evident in more commonly studied western populations, such as maternal birthplace (*Schooling et al., 2010*). For example, in the 'Children of 1997' birth cohort, a large cohort in Hong Kong, the associations of sugar-sweetened beverages (*Zhang et al., 2020*), breastfeeding (*Hui et al., 2018*), milk consumption frequency (*Lin et al., 2012*), sleep duration (*Wang et al., 2019*), and parental smoking (*Kwok et al., 2010*) with childhood and adolescent obesity have been examined, with some important differences detected. The associations for breastfeeding in Hong Kong are much more similar to those seen in randomized controlled trials (RCTs) than those typically seen in western settings (*Hui et al., 2018*). To take advantage of this unique setting, we conducted an environment- and epigenome-wide association study, to identify further potential drivers of obesity. We focused on puberty because it is an important stage involving a re-orientation from childhood priorities to adulthood (*Karlberg, 1989*). Exposures associated with obesity at puberty may be important for health in later life (*Richardson et al., 2020*). Considering that exposures related to obesity at the outset and at the end of puberty may be different, and the associations of the same exposure with obesity may vary by age, we conducted the EWAS at the onset and the end of puberty.

## Methods

### Participants

The study takes advantage of the 'Children of 1997' birth cohort, a large (*n* = 8327) population-representative Chinese cohort in Hong Kong (*Schooling et al., 2012*). The participants were originally recruited shortly after birth in April and May 1997 at all of the 49 Maternal and Child Health Centers (MCHCs) in Hong Kong, which provide free check-ups and immunizations. The study included 88% of births in the relevant period. A self-administered questionnaire in Chinese was used at baseline to collect information on family, education, birth characteristics, infant feeding, and second-hand smoke exposure. The initial study was designed to provide a short-term assessment of the effects of second-hand smoking and included follow-up via the MCHCs until 18 months. In 2005, funded by the Health and Health Services Research Fund (HHSRF) and Health and Medical Research Fund (HMRF) we extended the information on this cohort via record linkage to include infant characteristics, serious morbidity, childhood obesity, pubertal development, history of migration, and SEP; with regular updates on subsequent growth obtained from the Student Health Service, including annual height and weight measurements from age ~6 years, and in this study, we used height and weight at age ~11.5 years. In 2007, with support from The University of Hong Kong University Research Committee Strategic Research Theme of Public Health, we instituted a program to re-establish and maintain direct contact with the cohort through direct mailing (newsletters, birthday cards, and seasonal cards) and the mass media (press conference and a full-length television documentary). We have since conducted three questionnaires/telephone surveys and an in-person Biobank Clinical follow-up in one visit at age ~17.6 years (Phase 1 in 2013–2016 included 3460 people with mean age 17.5 years and Phase 2 in the second half of 2017 included 158 people at mean age 19.5 years) with their blood samples stored. In 2020 (at age ~23 years), we conducted a follow-up survey to obtain updated information on anthropometric measurements. The number of participants in each age is shown in *Figure 1—figure supplement 1*.

### Outcomes

At ~11.5 years, BMI was calculated from height and weight measurements records provided by the Student Health Service, the Department of Health. Waist–hip ratio (WHR) was calculated based on waist and hip circumference collected in Survey I conducted in 2008–2009. At ~17.6 years, in both phases of the Biobank Clinical follow-up, BMI was assessed by bio-electrical impedance analysis (BIA) with a Tanita segmental body composition monitor (Tanita BC-545, Tanita Co, Tokyo, Japan). Waist

and hip circumference measurements were made using a tape twice following a standard protocol by trained technicians and nurses. Given BMI has a more accepted cut-off value than WHR for children, we classified obesity as BMI ≥20.89 kg/m² for boys and BMI ≥21.20 kg/m² for girls at age 11.5; and BMI ≥24.73 kg/m² for men and BMI ≥24.85 kg/m² for women at age 17.6 (*Cole et al., 2000*). In the follow-up survey at ~23 years, questionnaires were sent to 700 participants randomly selected from those with blood samples available and with BMI below the 25th centile or above the 75th centile. The questionnaires were accompanied by clear instructions on anthropometric measurement and a tape measure (the same as used in the Biobank Clinical follow-up). In total, 308 participants replied and provided their waist, hip, height, and body weight.

## Assessment of DNA methylation

DNA methylation was conducted in 286 participants randomly selected from the 308 participants in the follow-up survey. DNA were extracted from buffy coat samples previously stored at −80° using EZI DNA blood kit (QIAGEN) with magnetic particle technology. DNA methylation was assessed using the Illumina Methylation EPIC Beadchip, which interrogated the methylation status of over 850,000 CpG sites. We conducted quality control using the 'ewastools' package (*Heiss and Just, 2018*), which included an evaluation of control metrics monitoring the various experimental steps, such as bisulfite conversion or staining and a sex check comparing actual sex to the records. After sample-level quality control, we excluded two samples that had a sex mismatch, so 286 samples (168 women and 118 men) were included in the analysis. We corrected for dye bias using RELIC (*Xu et al., 2017*), without normalization. At the probe level, we excluded non-CpG probes and probes located on the sex chromosomes; a total of 843,393 probes remained for analyses.

## Exposures and categorization

*Supplementary file 1* shows exposure categorization and data sources. After excluding exposures with missing values ≥50%, we included 123 exposures for BMI and 115 exposures for WHR at ~11.5 years, and 441 exposures for BMI and WHR at ~17.6 years using information from the original study, record linkage, the three surveys, that is Survey I (2008–2009), Survey II (2010–2012), and Survey III (2011–2012), as well as the Biobank Clinical follow-up. The exposures considered for obesity at ~11.5 years were classified into 12 categories, including baseline characteristics, SEP, family history, paternal information, maternal information, infant feeding and caring, diet (measured at Survey I at ~11.5 years old), health status (referring to physical health condition; details of the questions can be found in *Supplementary file 1*), parents' health status, physical activity, lifestyle, and home facilities and pets. The exposures at 17.6 years were classified into 16 categories: baseline characteristics, SEP, family history, paternal information, maternal information, infant feeding and caring, diet (measured at the Biobank Clinical follow-up at ~17.6 years), children's use of medications, children's health status, parent's health status, physical activity, home facilities and pets, moods and feelings, academic performance, sleep, and pubertal timing.

## Statistical analysis

In the EWAS, similar to genome-wide association studies (*Barrera-Gómez et al., 2017*), first we used univariable linear regression to assess associations of each of the exposures with the measures of obesity at ages ~11.5 and ~17.6 years. We conducted the analysis in people with both exposure and outcome available, specifically, in up to 7119 participants for BMI at ~11.5 years, 5691 participants for WHR at ~11.5 years, and 3618 participants for BMI and WHR at ~17.6 years; their baseline characteristics at different ages are shown in *Table 1*. We only considered exposures reaching Bonferroni-corrected significance (e.g. $p < 0.05/441 = 1.2 \times 10^{-4}$ for obesity at ~17.6 years) to account for multiple testing (*Curtin and Schulz, 1998*). Second, we used multivariable linear regression controlling for potential confounders (sex, housing type at birth, household income at birth, maternal second-hand smoking during pregnancy, maternal age at birth, maternal education, maternal birth place, and the interaction of maternal education with maternal birthplace [*Schooling et al., 2010*]) at age ~11.5 and ~17.6 years, and excluded exposures that had over 50% of change-in-estimates ratios (*Lee, 2014*). Third, to assess whether the associations differed by age, we checked the associations for the selected exposures from the earlier age groups (~11.5 and ~17.6 years) in the follow-up survey (*n* = 308) at age ~23 years and compared the direction of associations with those at earlier age groups (~11.5 and

**Table 1.** Baseline characteristics of Hong Kong's 'Children of 1997' birth cohort included in the environment-wide association study (EWAS) of obesity at 11.5, 17.6, and 23 years.

| Characteristics | | At ~11.5 **years** (n ≤ 7119) | At ~17.6 **years** (n = 3618) | At ~23 **years** (n = 308) |
|---|---|---|---|---|
| **Socio-demographics** | | | | |
| Sex | Boys | 3732 | 1825 | 123 |
| | Girls | 3387 | 1793 | 185 |
| Maternal birthplace | Not Hong Kong (migrant) | 2707 | 1509 | 128 |
| | Hong Kong | 4185 | 2089 | 179 |
| Highest maternal education | Grade 9 or below | 2873 | 1421 | 124 |
| | Grade 10–11 | 3060 | 1575 | 127 |
| | Grade 12 or above | 1064 | 600 | 57 |
| Household income per head in quintiles at recruitment | 1st quintile (HK$ 1746 ± 419) | 1260 | 599 | 45 |
| | 2nd quintile (HK$ 2853 ± 325) | 1288 | 647 | 49 |
| | 3rd quintile (HK$ 4365 ± 557) | 1269 | 646 | 58 |
| | 4th quintile (HK$ 6826 ± 883) | 1243 | 654 | 65 |
| | 5th quintile (HK$ 14,945 ± 15,620) | 1240 | 674 | 67 |
| Maternal age at delivery (mean [SD]) | | 30.2 (4.7) | 30.5 (4.6) | 30.8 (4.4) |
| **Anthropometrics** | | | | |
| Mean height in cm (SD) | Boys | 166.4 (7.5) | 171.8 (6.0) | 173.1 (6.3) |
| | Girls | 158.3 (5.9) | 159.5 (5.4) | 160.6 (5.2) |
| Mean weight in kg (SD) | Boys | 54.6 (11.3) | 62.7 (11.8) | 67.3 (15.0) |
| | Girls | 48.5 (8.0) | 52.6 (8.9) | 56.0 (30.5) |
| Mean BMI (SD) | Boys | 19.2 (3.7) | 21.1 (3.7) | 22.4 (4.8) |
| | Girls | 18.3 (3.1) | 20.6 (3.3) | 21.7 (11.5) |
| Mean WHR (SD) | Boys | 0.84 (0.07) | 0.79 (0.07) | 0.87 (0.07) |
| | Girls | 0.79 (0.06) | 0.75 (0.05) | 0.78 (0.07) |

~17.6 years). Associations with consistent directions of associations in earlier age groups (~11.5 or ~17.6 years) with those at ~23 years suggest a consistent association by age. Additionally, to account for the time lag between the age at which physical measurements were taken and age at exposure collection, we included a time difference variable in the adjusted models for BMI. Finally, exposures

that remained after controlling for confounders, were compared with the evidence from existing RCTs and Mendelian randomization (MR) studies, a study design which uses genetic variants as instrument and provides less confounded associations (*Smith and Ebrahim, 2003*).

In the epigenome-wide association study, considering the heteroscedasticity in methylation beta-values (*Du et al., 2010*), we used robust linear regression models to assess the epigenome-wide association of each CpG with BMI and WHR at age ~23 years. We adjusted for age at blood draw for DNA methylation, age at follow-up survey, sex, cell type proportion, methylation assay batches, maternal second-hand smoking during pregnancy, maternal education, maternal birthplace, and household income at birth. The significance was considered as $p < 1 \times 10^{-6}$, genome-wide significance ($5 \times 10^{-8}$) was not used given the relatively small sample size in the epigenome-wide association study. To estimate genomic inflation, we used a Bayesian method that estimates inflation more accurately in epigenome-wide association studies based on the empirical null distributions (*van Iterson et al., 2017*), implemented using the R package 'bacon'.

## Ethical approval

This study complies with the Declaration of Helsinki. Since our participants are children, informed (non-written) consent for the original survey and subsequent record linkage was obtained from the parents, next of kin, caretakers, or guardians (informants) on behalf of the participants by the informant agreeing and subsequently completing the questionnaire at enrollment, this manner of obtaining consent was approved by The University of Hong Kong Medical Faculty Ethics Committee over 20 years ago. Informed written consent for subsequent Surveys and in-person follow-up was obtained from a parent or guardian, or at ages 18+ years from the participant. Ethical approval for this study, including the follow-up survey at ~23 years and comprehensive health-related analyses, was obtained from the University of Hong Kong-Hospital Authority Hong Kong West Cluster, Joint Institutional Review Board, Hong Kong Special Administrative Region, China (reference numbers: UW13-367; UW19-367).

## Results

*Figures 1 and 2* show the association of each exposure with BMI and WHR at ~11.5 and ~17.6 years. At ~11.5 years, 18 associations with BMI and 19 associations with WHR remained after Bonferroni correction (*Figure 1*). Of these 18 associations with BMI, 14 associations with BMI remained after controlling for confounders, 13 showed significant association with obesity risk at 11.5 years, and 11 exposures had concordant direction of associations with BMI at ~23 years (*Table 2*). Of the 19 associations with WHR at ~11.5 years, 7 exposures remained after controlling for confounders, and 6 had the same direction of association with WHR at ~23 years (*Table 3*). At ~17.6 years, 37 associations with BMI and 19 with WHR remained after Bonferroni correction (*Figure 2*). Of these 37 associations with BMI, all remained after controlling for confounders, 27 exposures were associated with obesity risk, and 32 exposures had the same direction of association with BMI at ~23 years (*Table 4*). Of the 19 associations with WHR at ~17.6 years, 12 remained after controlling for confounders, and 7 had the same directions of association with WHR at ~23 years (*Table 5*).

Specifically, for obesity at ~11.5 years, we found sex (being male), higher birth weight, maternal second-hand smoking, higher parental weight, family history of diabetes, gestational diabetes, and more water consumption were associated with higher BMI, while being small for gestational age and spending more time having meals were associated with lower BMI at ~11.5 years (*Table 2*). Except for paternal diabetes which was not significant, the rest of exposures all showed consistent associations with obesity risk (*Table 2*). However, the associations for family history of diabetes and time spent on meals showed inconsistent directions of associations for BMI ~23 years (*Table 2*). Regarding WHR, the associations were generally consistent with those seen for BMI and showed consistent directions of associations with those at ~17.6 and ~23 years, except for maternal diabetes (*Table 3*).

For obesity at ~17.6 years, in addition to some shared factors, including sex, birth weight, parental weight, and maternal second-hand smoking, we found some aspects of diet (i.e. more artificially sweetened beverage [ASB], lower-sugar soy milk, reduced-fat/skim milk, Chinese herbal tea, Chinese tea, energy drinks, coffee, and fish consumptions), physical activity, health status (i.e. diabetes, growth problem and snoring), earlier puberty and binge eating associated with higher BMI at ~17.6 years

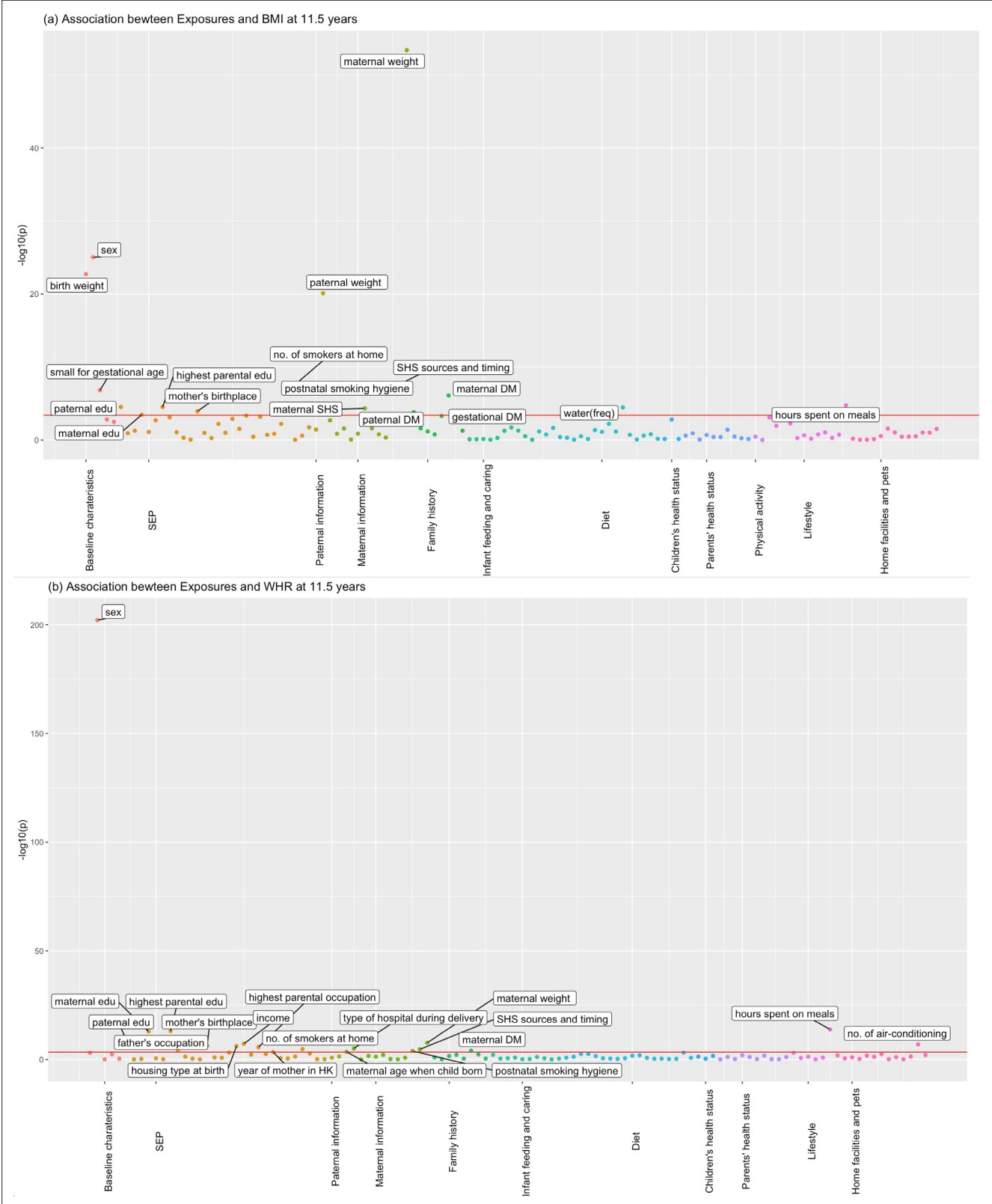

**Figure 1.** Associations of all exposures with body mass index (BMI) and waist–hip ratio (WHR) at age ~11.5 years in the univariable regression in Hong Kong's 'Children of 1997' birth cohort. (**a**) Associations of all exposures with body mass index (BMI) at age ~11.5 years in the univariable regression in Hong Kong's 'Children of 1997' birth cohort. (**b**) Associations of all exposures with waist–hip ratio (WHR) at age ~11.5 years in the univariable regression in Hong Kong's 'Children of 1997' birth cohort. DM, diabetes; edu, education; ASB, artificially sweetened beverages; PA, physical activity; SHS, second-

*Figure 1 continued on next page*

*Figure 1 continued*

hand smoking; freq, frequency. In total, we included 123 exposures for BMI at 11.5 years and 115 exposures for WHR at 11.5 years. The cut-off lines indicate Bonferroni-corrected p thresholds (p < 0.05/123 = 4.07 × 10⁻⁴ for BMI, p < 0.05/115 = 4.35 × 10⁻⁴ for WHR).

The online version of this article includes the following source data and figure supplement(s) for figure 1:

**Source data 1.** The associations for depicting *Figure 1a* in the study.

**Source data 2.** The associations for depicting *Figure 1b* in the study.

**Figure supplement 1.** Number of participants in the environment-wide (EWAS) and epigenome-wide association study of adiposity in 'Children of 1997' birth cohort.

(*Table 4*). Being a twin, sweets consumption, chocolate consumption, eating before sleep, and having bad dreams were associated with lower BMI at ~17.6 years (*Table 4*). Regarding the association with obesity risk at 17.6 years, birth weight, being a twin, maternal second-hand smoking, physical activity, and energy drinks intake were not significant but the direction of association was consistent (*Table 4*). In addition, most exposures had the same direction of association at ~23 years. However, the association of chocolate consumption with BMI at ~23 years was in the other direction. Regarding WHR, the associations were generally consistent with those seen for BMI. Sex, drinking ASB, children's health status, and coughing or snoring during sleep were also related to WHR at ~17.6 and ~23 years (*Table 5*). The associations of selected exposures with BMI at ~11.5 and ~17.6 years were similar after adjusting for the time difference between age at anthropometric measurements and age of exposure collection.

Regarding the comparison with RCTs and MR studies, we found RCTs on dark chocolate consumption (*Kord-Varkaneh et al., 2019*), water consumption promotion (*Muckelbauer et al., 2009*), and physical activity (*Bleich et al., 2018*), and MR studies related to drinking coffee (*Nordestgaard et al., 2015*), dairy intake (*Huang et al., 2018b*), binge eating (*Reed et al., 2017*), physical activity (*Carrasquilla et al., 2022*), snoring (*Campos et al., 2020*), puberty (*Gill et al., 2018*; *Bell et al., 2018*), birth weight (*Zanetti et al., 2018*), and maternal obesity (*Bond et al., 2022*; *Richmond et al., 2017*; *Table 6*). The available evidence from both RCTs and MR studies show that more physical activity lowers BMI (*Bleich et al., 2018*), and MR studies also suggest that dairy intake (*Yang et al., 2017*; *Huang et al., 2018b*) and binge eating (*Reed et al., 2017*) are associated with higher BMI.

In the epigenome-wide association study of 286 participants we identified 17 CpGs for BMI at ~23 years in the genes *RBM16*, *SCN2B*, *SLC24A4*, *TECPR2*, *KSR1*, *RPTOR*, *GTF3C3*, *ZNF827*, *TXNDC15*, *C2*, and *RPS6KA2* and 17 for WHR in the genes *LANCL2*, *C6orf195*, *MIR4535*, *CTRL*, *LYRM9*, *DCDC2*, *DIRC3*, *RPS6KA2*, *LPP*, *NFIC*, *MIR7641-2*, *ZNF141*, *RNF213*, and *OPA3* (*Tables 7 and 8*; *Figures 3 and 4*). cg14630200 in *RPS6KA2* was a shared CpG for both BMI and WHR. The genetic inflation factors (lambda) were 1.006 and 1.005, respectively.

## Discussion

In this environment- and epigenome-wide association study, we systematically examined associations of over 400 exposures with obesity in a unique Chinese birth cohort, as well as the association of DNA methylation with obesity. Building on the previous studies in this birth cohort (*Zhang et al., 2020*; *Hui et al., 2018*; *Lin et al., 2012*; *Kwok et al., 2010*), we not only confirmed established risk factors, such as maternal second-hand smoking (*Wang et al., 2014*), but also added by identifying novel exposures not reported in previous EWAS in western settings (*Vrijheid et al., 2020*; *Uche et al., 2020*), such as consumption of ASB and soymilk. The comparison with RCTs or MR studies support a role of higher birth weight, dairy intake, binge eating, and possibly earlier puberty in obesity. We also identified several CpGs related to BMI and WHR in young Chinese, as reported in other populations (*Kvaløy et al., 2018*).

Our study found that maternal second-hand smoking was consistently associated with obesity at different ages, which is consistent with the concerns repeatedly raised in previous studies (*Kwok et al., 2010*; *Wang et al., 2014*), and adds support to the policy of banning smoking cigarettes and alternative smoking products in all indoor areas including workplaces and public places, as well as certain outdoor areas, such as open areas of schools, leisure facilities, bathing beaches, and public transport facilities in Hong Kong (*WSC, 2022*). Maternal weight is another maternal factor related to

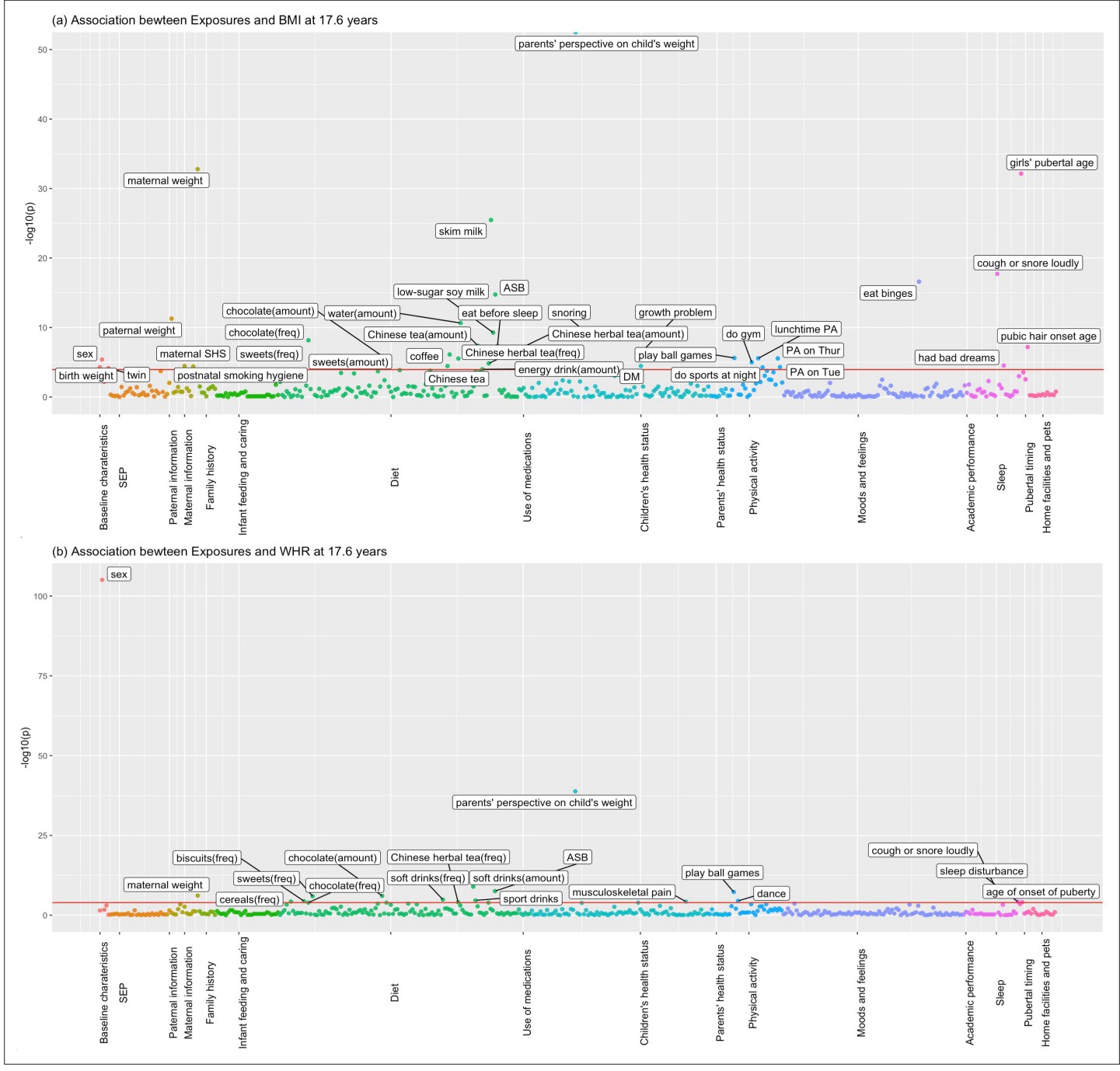

**Figure 2.** Associations of all exposures with body mass index (BMI) and waist–hip ratio (WHR) at age ~17.6 years in the univariable regression in 3618 participants of Hong Kong's 'Children of 1997' birth cohort in the Biobank Clinical follow-up. (**a**) Associations of all exposures with body mass index (BMI) at age ~17.6 years in the univariable regression in 3618 participants of Hong Kong's 'Children of 1997' birth cohort in the Biobank Clinical follow-up. (**b**) Associations of all exposures with waist–hip ratio (WHR) at age ~17.6 years in the univariable regression in 3618 participants of Hong Kong's 'Children of 1997' birth cohort in the Biobank Clinical follow-up. DM, diabetes; ASB, artificially sweetened beverages; PA, physical activity; SHS, second-hand smoking; freq, frequency. The cut-off lines indicate Bonferroni-corrected p thresholds (p < 0.05/441 exposures = $1.2 \times 10^{-4}$ for BMI and WHR).

The online version of this article includes the following source data for figure 2:

**Source data 1.** The associations for depicting *Figure 2a* in the study.

**Source data 2.** The associations for depicting *Figure 2b* in the study.

**Table 2.** Associations of selected exposures with body mass index (BMI) and obesity risk at ~11.5 years, and associations of those exposures with BMI at ~17.6 years and with BMI at ~23 years in participants of Hong Kong's 'Children of 1997' birth cohort.

| Group names | Variable description | With BMI at ~11.5 years* | | With obesity risk at ~11.5 years* | With BMI at ~17.6 years[†] | | With BMI at ~23 years[†] | |
|---|---|---|---|---|---|---|---|---|
| | | Beta | p value | OR (95% CI) | Beta | p value | Beta | p value |
| Baseline characteristics | Sex | 0.85 | 5.65E−22 | 2.06 (1.82, 2.32) | 0.23 | 1.99E−03 | 0.25 | 8.44E−01 |
| Baseline characteristics | Birth weight | 0.40 | 5.40E−15 | 1.23 (1.15, 1.32) | 0.56 | 7.84E−06 | 0.61 | 3.78E−01 |
| Baseline characteristics | Small for gestational age: birth weight <10% by sex and gestational week distribution in singletons | −0.68 | 9.93E−06 | 0.71 (0.56, 0.89) | −0.42 | 5.65E−02 | −0.97 | 6.46E−01 |
| SEP | No. of smokers at home | 0.15 | 2.83E−02 | 1.15 (1.06, 1.25) | 0.20 | 4.26E−02 | 1.07 | 3.06E−01 |
| Paternal information | Paternal weight | 0.05 | 1.58E−18 | 1.03 (1.02, 1.04) | 0.04 | 1.73E−09 | 0.04 | 5.68E−01 |
| Maternal information | Maternal weight | 0.09 | 1.84E−49 | 1.06 (1.05, 1.07) | 0.09 | 1.30E−28 | 0.18 | 5.23E−02 |
| Maternal information | Postnatal smoking hygiene (i.e. second-hand smoking by timing [pre- and/or postnatal]) | 0.10 | 7.79E−05 | 1.06 (1.02, 1.10) | 0.14 | 2.51E−04 | 0.36 | 3.67E−01 |
| Maternal information | Second-hand smoke by sources and timing | 0.08 | 2.69E−05 | 1.05 (1.02, 1.08) | 0.10 | 3.31E−04 | 0.23 | 4.54E−01 |
| Maternal information | Mother exposed to second-hand smoke during pregnancy | 0.15 | 4.07E−03 | 1.08 (1.01, 1.16) | 0.23 | 9.70E−05 | 0.32 | 5.85E−01 |
| Family history | Paternal diabetes | 0.69 | 5.82E−03 | 1.31 (0.93, 1.86) | 0.31 | 2.10E−01 | 8.74 | 2.08E−04 |
| Family history | Maternal diabetes | 1.45 | 2.47E−05 | 2.21 (1.42, 3.44) | 0.36 | 2.23E−01 | −0.89 | 7.67E−01 |
| Family history | Gestational diabetes | 0.68 | 2.28E−04 | 1.51 (1.18, 1.93) | 0.57 | 1.84E−01 | −0.83 | 9.37E−01 |
| Diet | Water: frequency of consumption in the last week | 0.32 | 9.45E−06 | 1.20 (1.06, 1.36) | 0.37 | 1.89E−04 | 1.27 | 2.38E−01 |
| Lifestyle | Having meals: hours spent yesterday | −0.40 | 2.06E−03 | 0.80 (0.65, 0.99) | −0.42 | 1.65E−02 | 1.71 | 3.59E−01 |

*Association of selected exposures with BMI and obesity risk at age ~11.5 years in up to 7119 participants after controlling for confounders.

[†]Association of selected exposures for BMI at age ~11.5 years with BMI at age ~17.6 years (n = 3618) and ~23 years (n = 308) after controlling for confounders.

higher BMI and WHR consistently at different ages before adulthood. However, recent MR studies do not support a role of maternal overweight in offspring obesity (*Bond et al., 2022*; *Richmond et al., 2017*). Gestational diabetes was also identified to be associated with obesity at ~11.5 years, and the positive association remained for obesity at ~17.6 and ~23 years. It would be worthwhile to test its role in MR studies.

Regarding dietary factors, as the dietary assessments were more comprehensively conducted in the Biobank Clinical follow-up, the identified dietary factors were mainly for obesity at ~17.6 years. Interestingly, we found that children who consume more ASB have higher BMI, which is consistent with meta-analyses of cohort studies (*Qin et al., 2020*; *Rousham et al., 2022*). Consistent with a previous study in this birth cohort (*Zhang et al., 2020*), we did not find an association of sugar-sweetened beverages with obesity. The different associations for ASB and sugar-sweetened beverages might be because few consumed sugar-sweetened beverages regularly (6.8% consumed daily) (*Zhang et al., 2020*), while many consumed ASB (43% participants reported consumption). Consistent with our EWAS, an EWAS in the US also found that consumption of aspartame, a synthetic non-nutritive sweetener, was positively associated with abdominal obesity (*Wulaningsih et al., 2017*). ASB intake may induce appetite for similar sweet foods, leading to excess energy intake (*Mattes and Popkin, 2009*). The consistency across settings suggests this association is less likely to be confounded. However, whether it can be used as a target of intervention needs to be tested in a randomized, placebo-controlled trial (*Rousham et al., 2022*).

**Table 3.** Associations of selected exposures with waist–hip ratio (WHR) at ~11.5 years, and associations of those exposures with WHR at ~17.6 years and with WHR at ~23 years in participants of Hong Kong's 'Children of 1997' birth cohort.

| Group names | Variable description | With WHR at ~11.5 years[*] | | With WHR at ~17.6 years[†] | | With WHR at ~23 years[†] | |
|---|---|---|---|---|---|---|---|
| | | Beta | p value | Beta | p value | Beta | p value |
| Baseline characteristics | Sex | 0.05 | 6.58E−171 | 0.04 | 1.28E−88 | 0.082 | 4.30E−18 |
| SEP | Housing type at birth (home ownership) | −0.01 | 2.70E−02 | −0.001 | 8.51E−01 | −0.01 | 4.20E−01 |
| SEP | Maternal education level | −0.01 | 4.90E−05 | −0.004 | 2.03E−01 | −0.001 | 9.02E−01 |
| SEP | Mother's birthplace | −0.01 | 1.18E−01 | −0.01 | 2.76E−01 | −0.006 | 7.97E−01 |
| Maternal information | Maternal weight | 0.00 | 7.98E−10 | 0.001 | 1.13E−06 | 0.001 | 6.58E−02 |
| Family history | Maternal diabetes | 0.03 | 2.30E−05 | 0.01 | 1.27E−01 | −0.017 | 7.82E−01 |
| Lifestyle | Having meals: hours spent yesterday | −0.01 | 8.15E−09 | −4.73E−03 | 1.30E−01 | −0.002 | 8.11E−01 |

[*]Association of selected exposures with WHR at age ~11.5 years in up to 7119 participants after controlling for confounders.

[†]Association of selected exposures for WHR at age ~11.5 years with BMI at age ~17.6 years (n = 3618) and ~23 years (n = 308) after controlling for confounders.

Another interesting finding is that milk consumption was not related to obesity at ~11.5 years, while reduced-fat/skim milk consumption was associated with higher BMI at ~17.6 years, with a consistent direction of associations for BMI at ~23 years. Our findings are consistent with a previous study in this cohort, which showed milk consumption frequency was not associated with BMI at 13 years (*Lin et al., 2012*), however, the previous study did not assess the specific type of milk. Our findings are different from a cross-sectional study in Portugal which shows more skimmed or semi-skimmed milk consumption was associated with lower abdominal obesity (*Abreu et al., 2014*). An explanation is that the observed associations of reduced-fat or skim milk with higher BMI could be due to increased muscle mass rather than body fat mass, or residual confounding by SEP. Alternatively, it might be due to reverse causality as young people with higher BMI might be more motivated to consume a specific diet. Interestingly, our findings are more consistent with an MR study suggesting genetically predicted higher dairy intake was associated with higher BMI (*Huang et al., 2018b*).

We also found tea or coffee consumption were associated with higher BMI at ~17.6 years. RCTs of coffee or tea consumption are scarce among children and adolescents because they require long-term adherence. MR studies do not suggest that coffee consumption affects obesity (*Nordestgaard et al., 2015*; *Cornelis and Munafo, 2018*), so the observed association might be due to confounding or chance. Similarly, the associations of chocolate and sweets intake with lower BMI at ~17.6 years, as well as physical activity with higher BMI at ~17.6 years are not consistent with RCTs of dark chocolate consumption (*Kord-Varkaneh et al., 2019*) and physical activity (*Bleich et al., 2018*), or MR studies of physical activity (*Carrasquilla et al., 2022*), and might be due to confounding or reverse causality.

Echoing the increasing attention to the role of mood and emotion in obesity control (*Cardel et al., 2020*), we found that binge eating was associated with higher BMI at ~17.6 years, with a consistent direction of association in the follow-up, consistent with an MR study (*Reed et al., 2017*). Our findings are also in line with the National Institute for Health and Care Excellence (NICE) guidance which also included binge eating in the consideration of children's weight management (*NICE, 2013*). The underlying mechanism has not been clarified, but in general mental wellbeing may be linked to obesity via the neurohormonal weight control network concerning the hypothalamus (*Sharma and Kavuru, 2010*; *Spiegel et al., 2009*) as well as via psychosocial factors, lifestyle and behaviour.

As regards lifestyle, consistent with previous observational studies (*Wang et al., 2019*), we found sleep might play a role in childhood obesity at ~17.6 years. Despite a lack of RCTs, MR findings suggest sleep deprivation may be a causal factor for obesity (*Wang et al., 2019*; *Dashti and Ordovás, 2021*). Our study also shows coughing or snoring at night had a positive association with childhood BMI and WHR, which has not been identified in previous EWAS (*Vrijheid et al., 2020*; *Uche et al., 2020*). MR studies suggest genetically predicted BMI is positively associated with snoring (*Campos*

**Table 4.** Associations of selected exposures with body mass index (BMI) and obesity risk at ~17.6 years, and associations of those exposures with BMI at ~23 years in participants from Hong Kong's 'Children of 1997' birth cohort.

| Group names | Variable description | With BMI at ~17.6 years* | | With obesity risk at ~17.6 years* | With BMI at ~23 years† | |
|---|---|---|---|---|---|---|
| | | Beta | p value | OR (95% CI) | Beta | p value |
| Baseline characteristics | Sex | 0.56 | 7.84E−06 | 1.72 (1.38, 2.14) | 0.25 | 8.44E−01 |
| Baseline characteristics | Birth weight | 0.23 | 1.99E−03 | 1.1 (0.97, 1.25) | 0.61 | 3.78E−01 |
| Baseline characteristics | Twin | −1.77 | 1.05E−03 | 0.17 (0.02, 1.27) | −3.73 | 5.28E−01 |
| Paternal information | Paternal weight | 0.04 | 1.73E−09 | 1.03 (1.02, 1.04) | 0.04 | 5.68E−01 |
| Maternal information | Maternal weight | 0.09 | 1.30E−28 | 1.05 (1.03, 1.06) | 0.18 | 5.23E−02 |
| Maternal information | Mother exposed to second-hand smoke during pregnancy | 0.23 | 9.70E−05 | 1.12 (0.99, 1.27) | 0.02 | 9.78E−01 |
| Maternal information | Postnatal smoking hygiene (i.e. second-hand smoking by timing [pre- and/or postnatal]) | 0.09 | 6.88E−02 | 1.72 (1.38, 2.14) | 0.78 | 1.57E−01 |
| Diet | Chocolate: frequency of consumption in the past week | −0.34 | 9.02E−06 | 0.73 (0.63, 0.84) | 0.29 | 6.92E−01 |
| Diet | Chocolate: no. of servings/time | −0.19 | 1.09E−04 | 0.83 (0.73, 0.95) | 0.08 | 8.69E−01 |
| Diet | Sweets: frequency of consumption in the past week | −0.25 | 4.18E−04 | 0.83 (0.73, 0.95) | −0.14 | 8.25E−01 |
| Diet | Sweets: no. of servings/time | −0.17 | 5.42E−05 | 0.85 (0.77, 0.94) | −0.17 | 6.85E−01 |
| Diet | Chinese tea: frequency of consumption in the past week | 0.27 | 1.69E−05 | 1.25 (1.13, 1.38) | 0.29 | 6.30E−01 |
| Diet | Chinese tea: no. of cup/time | 0.17 | 2.08E−07 | 1.14 (1.08, 1.2) | 0.18 | 5.74E−01 |
| Diet | Chinese herbal tea: frequency of consumption in the past week | 0.32 | 3.16E−04 | 1.18 (1.02, 1.36) | 1.11 | 1.82E−01 |
| Diet | Chinese herbal tea: no. of cup/time | 0.12 | 9.17E−04 | 1.13 (1.05, 1.22) | 0.26 | 5.01E−01 |
| Diet | Coffee: frequency of consumption in the past week | 0.30 | 2.90E−05 | 1.17 (1.04, 1.32) | 0.38 | 6.05E−01 |
| Diet | Water: no. of cup/time | 0.25 | 1.07E−09 | 1.13 (1.05, 1.22) | 0.51 | 1.97E−01 |
| Diet | Energy drinks: no. of cup/time | 0.14 | 1.21E−04 | 1.13 (0.99, 1.3) | 0.66 | 4.84E−01 |
| Diet | When consume milk, how often reduced-fat/skim milk | 0.60 | 2.56E−24 | 0.7 (0.63, 0.77) | 1.64 | 5.35E−03 |
| Diet | When consume soy milk/flavored drinks, how often lower sugar | 0.48 | 5.14E−11 | 0.73 (0.64, 0.82) | 0.78 | 2.55E−01 |
| Diet | When consume soft drinks, how often diet/artificially sweetened drinks | 0.50 | 1.75E−10 | 0.7 (0.62, 0.8) | 0.62 | 3.61E−01 |
| Diet | Eating: 1 hr before sleeping | −0.76 | 2.85E−06 | 0.65 (0.47, 0.9) | −1.04 | 5.04E−01 |
| Children's Health | Parents' questionnaire: do you consider your child now to be: very thin/underweight; a little bit thin/underweight; about the right weight; a little bit fat/overweight; very fat/overweight; don't know | 2.51 | 0.00E+00 | 7.45 (6.09, 9.12) | 2.13 | 4.12E−04 |
| Children's Health | Snoring in the past 4 weeks (child) | 0.76 | 9.72E−08 | 0.51 (0.4, 0.65) | 1.30 | 3.18E−01 |
| Children's Health | Growth problem (child) | 0.36 | 7.41E−08 | 1.5 (1.37, 1.64) | −0.02 | 9.76E−01 |
| Children's Health | Diabetes (child) | 0.32 | 4.62E−04 | 1.29 (1.15, 1.45) | 0.01 | 9.94E−01 |
| Physical activity | Ball games: no. of times you done past week | 0.22 | 4.04E−03 | 1.03 (0.92, 1.17) | 1.29 | 9.89E−02 |
| Physical activity | Gymnasium: no. of times you done past week | 0.44 | 3.10E−05 | 1.14 (0.97, 1.34) | 0.25 | 8.18E−01 |
| Physical activity | What did you normally do at lunch in the past 7 days | 0.34 | 5.50E−05 | 1.22 (1.06, 1.39) | −0.22 | 8.12E−01 |
| Physical activity | How many nights did you do sports in the past 7 days | 0.16 | 1.13E−02 | 0.98 (0.88, 1.08) | 0.65 | 3.43E−01 |

*Table 4 continued on next page*

*Table 4 continued*

| Group names | Variable description | With BMI at ~17.6 years[*] | | With obesity risk at ~17.6 years[*] | With BMI at ~23 years[†] | |
|---|---|---|---|---|---|---|
| | | Beta | p value | OR (95% CI) | Beta | p value |
| Physical activity | Tuesday: how often you did physical activity | 0.18 | 2.15E−03 | 1.01 (0.92, 1.11) | −0.04 | 9.48E−01 |
| Physical activity | Thursday: how often you did physical activity | 0.21 | 2.86E−04 | 1.08 (0.98, 1.2) | 1.50 | 9.65E−03 |
| Moods and feelings | Has he/she gone on eating binges | 3.22 | 5.23E−15 | 5.19 (3.18, 8.46) | 3.37 | 3.06E−01 |
| Sleep | Cough or snore loudly | 0.67 | 1.62E−13 | 1.43 (1.25, 1.63) | 1.15 | 2.15E−01 |
| Sleep | Have bad dreams | −0.27 | 1.87E−03 | 0.77 (0.64, 0.91) | −0.99 | 2.31E−01 |
| Pubertal timing | Age of onset of pubic hair | −0.56 | 2.41E−10 | 0.78 (0.67, 0.92) | −0.85 | 1.24E−03 |
| Pubertal timing | Age of menarche (girls) | −0.81 | 2.46E−27 | 0.54 (0.45, 0.64) | −1.11 | 2.18E−01 |

[*]Association of selected exposures with BMI and obesity risk at age ~17.6 years in 3618 participants after controlling for confounders.
[†]Association of selected exposures for BMI at age ~17.6 years with BMI at ~23 years (*n* = 308) after controlling for confounders.

*et al., 2020*), while the association of snoring with BMI is less clear. As such, we cannot exclude the possibility of reverse causality in our observation.

Consistent with previous studies (*Lai et al., 2021*), we found that earlier pubertal age for girls was related to higher BMI at ~17.6 years. Consistently, a previous study in this birth cohort suggested that maternal age at puberty was associated with offspring BMI at puberty (*Lai et al., 2016*). However, the findings from MR studies are controversial, with evidence showing genetically predicted earlier age at puberty related to higher BMI (*Gill et al., 2018*) while another MR study showing puberty timing has a small influence on BMI (*Bell et al., 2018*). Meanwhile, we cannot exclude a relation in the other direction (*Chen et al., 2019*); clarifying the bi-directional association would be worthwhile in future studies.

**Table 5.** Associations of selected exposures with waist–hip ratio (WHR) at ~17.6 years, and associations of those exposures with WHR at ~23 years in participants from Hong Kong's 'Children of 1997' birth cohort.

| Group names | Variable description | With WHR at ~17.6 years[*] | | With WHR at ~23 years[†] | |
|---|---|---|---|---|---|
| | | Beta | p value | Beta | p value |
| Baseline characteristics | Sex | 0.045 | 1.28E−88 | 0.083 | 5.71E−18 |
| Maternal information | Maternal weight | 0.001 | 1.13E−06 | 0.001 | 9.38E−02 |
| SEP | Housing type at birth (home ownership) | −0.006 | 2.70E−02 | −0.006 | 6.10E−01 |
| Diet | Biscuits: frequency of consumption in the past week | −0.003 | 2.60E−02 | 0.000 | 9.36E−01 |
| Diet | Chocolate: frequency of consumption in the past week | −0.004 | 1.86E−03 | −0.003 | 5.20E−01 |
| Diet | Chocolate: no. of servings/time | −0.003 | 5.03E−05 | 0.000 | 9.11E−01 |
| Diet | Sweets: frequency of consumption in the past week | −0.003 | 1.07E−02 | 0.000 | 9.80E−01 |
| Diet | When consume soft drinks, how often diet/artificially sweetened drinks | 0.005 | 1.04E−04 | −0.008 | 9.36E−02 |
| Children's Health | Parents' questionnaire: do you consider your child now to be: very thin/underweight; a little bit thin/underweight; about the right weight; a little bit fat/overweight; very fat/overweight; don't know | 0.021 | 3.55E−59 | 0.015 | 3.02E−04 |
| Children's Health | Whether experienced musculoskeletal pain in past 4 weeks | 0.006 | 1.57E−02 | −0.001 | 8.94E−01 |
| Sleep | Cough or snore loudly | 0.007 | 2.30E−06 | 0.013 | 4.76E−02 |
| Sleep | Sleep disturbance (child) | 0.005 | 1.93E−03 | 0.003 | 6.28E−01 |

[*]Association of selected exposures with WHR at age ~17.6 years in 3618 participants after controlling for confounders.
[†]Association of selected exposures for WHR at age ~17.6 years with WHR at ~23 years (*n* = 308) after controlling for confounders.

**Table 6.** Evidence from published systematic reviews, randomized controlled trials (RCTs) and Mendelian randomization (MR) studies regarding the role of exposures selected in our environment-wide association study (EWAS) in obesity.

| Exposure | Published RCTs studies | Published MR studies |
|---|---|---|
| Water consumption | Intervention on promoting water consumption showed no effect on body mass index (BMI; *Muckelbauer et al., 2009*). | NA |
| Reduced-fat/skim milk consumption | NA | MR study on different types of milk consumption (i.e. reduced-fat, skimmed, reduced-sugar milk) among children is lacking. Nevertheless, two MR studies suggested higher dairy intake was associated with higher adult's BMI (*Huang et al., 2018b*; *Yang et al., 2017*). |
| Coffee consumption | NA | Genetically predicted more coffee intake was not associated with obesity, BMI, or waist circumference in two large adult population cohorts (*Nordestgaard et al., 2015*). Also, most MR studies do not support a role of caffeine consumption on BMI or waist circumstance (*Nordestgaard et al., 2015*; *Cornelis and Munafo, 2018*). |
| Chocolate consumption | Meta-analysis of RCTs did not support a significant effect of cocoa/dark chocolate supplementation on body weight or BMI (*Kord-Varkaneh et al., 2019*). | NA |
| Diabetes | NA | MR study supported genetic predisposition to higher childhood BMI was associated with risk of type 2 diabetes (*Geng et al., 2018*). |
| Binge eating | NA | The MR study suggests a bi-directional association, that is, more binge eating and overeating are associated with higher BMI in later life, and higher children's BMI is associated with more binge eating (*Reed et al., 2017*). |
| Physical exercises | A systematic review shows school based RCTs targeting physical activity, or physical activity combined with diet interventions, were effective in reducing BMI among children (*Bleich et al., 2018*). | Genetically predicted more physical activity was associated with lower BMI (*Carrasquilla et al., 2022*). |
| Snoring | NA | MR study suggests genetically predicted BMI is related to snoring (*Campos et al., 2020*). |
| Pubertal timing | NA | Genetically predicted earlier age at puberty was related to higher BMI (*Gill et al., 2018*), but pleiotropy might exist (*Gill et al., 2018*). The association attenuated towards the null after controlling for prior BMI (*Bell et al., 2018*). It is also possible that genetically predicted BMI was associated with earlier age at puberty (*Chen et al., 2019*). |
| Birth weight | NA | MR analyses indicated positive causal associations of birthweight with BMI in the UKB (*Zanetti et al., 2018*). |
| Maternal adiposity during pregnancy | NA | Using BMI polygenic risk scores calculated from maternal non-transmitted alleles, previous MR studies using mother–offspring pairs from two large UK cohorts did not support causal associations between maternal pre/early pregnancy BMI and offspring adolescent adiposity (*Bond et al., 2022*; *Richmond et al., 2017*). |

In the epigenome-wide association study, we found DNA methylation at *RPS6KA2* was associated with both BMI and WHR, consistent with the previous epigenome-wide association studies of obesity in different populations (*Kvaløy et al., 2018*). Our study also identified several other genes, such as *ZNF827*, *MIR7641-2*, *RAPTOR*, *KSR1*, *GTF3C3*, and *NFIC*, whose role in obesity or obesity-related disorders has been consistently shown in previous studies. For example, *ZNF827*, *MIR7641-2*, and *RAPTOR* have been reported to be related to obesity (*Huang et al., 2015*; *Dong et al., 2016*) and/or overweight (*Morris et al., 2015*). *KSR1* has been reported to be related to the regulation of glucose homeostasis (*Klutho et al., 2011*), and *GTF3C3*- to obesity-related dysglycaemia (*Andrade et al., 2021*). *NFIC*, which encodes nuclear factor I-C, regulates adipocyte differentiation (*Zhou et al., 2017*). *Opa3*, a novel regulator of mitochondrial function, controls thermogenesis and abdominal fat mass (*Wells et al., 2012*). The consistency of our study with other studies in different settings with different confounding structures suggests these association are less likely to be a product of confounding.

**Table 7.** Associations of CpGs with body mass index (BMI) at age ~23 years in 286 participants from Hong Kong's 'Children of 1997' birth cohort in the follow-up survey.

| CpG | Gene | Beta | p value |
|------|------|------|---------|
| cg00379266 | *RBM16* | −69.7 | 9.8E−07 |
| cg01525498 | *RPTOR* | −40.2 | 4.6E−10 |
| cg02200725 | - | −18.9 | 4.2E−07 |
| cg04563671 | *SCN2B* | 14.8 | 1.6E−07 |
| cg07243141 | - | 24.6 | 9.0E−07 |
| cg08205320 | - | −21.9 | 2.6E−08 |
| cg09639369 | *SLC24A4* | −15.7 | 5.7E−07 |
| cg11937362 | - | 109 | 9.5E−07 |
| cg12484266 | *C2* | 83 | 5.7E−09 |
| cg14630200 | RPS6KA2 | −45.1 | 7.1E−12 |
| cg14850190 | KSR1 | 103.2 | 4.3E−07 |
| cg18681028 | - | −48.5 | 7.6E−16 |
| cg18893311 | - | 30.8 | 7.5E−08 |
| cg22197830 | TXNDC15 | −68.8 | 4.9E−07 |
| cg25356423 | ZNF827 | −14 | 1.6E−07 |
| cg25903177 | - | −31.4 | 5.8E−07 |
| cg26347007 | GTF3C3 | −45.8 | 8.0E−07 |

CpGs reaching significance of $p < 1 \times 10^{-6}$ were shown in the table.

## Strengths and limitations

To our knowledge, this study is the first study comprehensively assessing environmental factors related to obesity at the outset and at end of puberty in Asians, including some exposures specifically relevant in Asians, such as soymilk intake. We also replicated the associations in a follow-up survey and compared our findings with those from studies of different designs. Nevertheless, several limitations exist. First, the sample size for the EWAS and epigenome-wide association study is relatively small. Replication in a larger study is needed. The evidence from RCTs and MR is mainly in adults, which restricted the comparison with our findings. For better comparison, evidence from RCTs and MR conducted at similar ages are needed. Second, misclassification is possible for the exposures, which typically biases towards the null (*Rothman, 2008*). The use of questionnaires to ascertain exposures is prone to recall bias and social desirability bias, however, some previous studies within this cohort have suggested accurate reporting (*Kwok et al., 2010*). To minimize these possible biases, exposures measurements were collected using standard protocols and equipment with clear instructions. After accounting for the time difference between age at anthropometric measurements and age of exposure collection, we also obtained similar results. Third, we obtained similar results for BMI and obesity risk, however, given the lack of high-quality evidence about the cut-off values for waist circumference and waist-to-hip ratio in Asian children and adolescents, we did not perform logistic regression on central obesity risk. Fourth, the inconsistency between some of our findings and previous studies, such as chocolate, sweets, tea, and coffee consumption, should be interpreted cautiously. It may not only reflect differences between the West and China (Hong Kong), but also may be due to changes in structural socioeconomic and environmental factors, as well as changes in living environment, family relationships, social and community networks, housing and the care environment. Fifth, we only collected blood samples at the Biobank Clinical follow-up (age ~17.6 years), so we only conducted the epigenome-wide association study for DNA methylation at ~17.6 years. It would be worthwhile to examine the association of DNA methylation at different ages with obesity. Finally, although we controlled for several confounders in the multivariable analysis, residual confounding may still exist

**Table 8.** Associations of CpGs with waist–hip ratio (WHR) at age ~23 years in 286 participants from Hong Kong's 'Children of 1997' birth cohort in the follow-up survey.

| CpG | Gene | Beta | p value |
|---|---|---|---|
| cg00952960 | LANCL2 | −0.54 | 3.0E−11 |
| cg04405211 | C6orf195 | 0.35 | 9.7E−07 |
| cg05059349 | - | 0.26 | 3.7E−07 |
| cg07416310 | DCDC2 | −0.45 | 8.8E−08 |
| cg09422806 | NFIC | 0.5 | 3.4E−08 |
| cg09684856 | DIRC3 | −0.79 | 1.4E−07 |
| cg09907395 | RNF213 | 1.09 | 2.2E−07 |
| cg11344771 | LPP | −0.79 | 1.0E−08 |
| cg14630200 | RPS6KA2 | −0.46 | 5.4E−07 |
| cg17007012 | MIR4535 | −0.35 | 6.6E−11 |
| cg17043713 | ZNF141 | 0.55 | 8.1E−08 |
| cg17769811 | CTRL | −0.76 | 1.3E−09 |
| cg19481727 | MIR7641-2 | −0.36 | 5.4E−07 |
| cg19579176 | - | −0.35 | 9.9E−07 |
| cg22364817 | - | −1.46 | 6.7E−07 |
| cg25097095 | LYRM9 | −0.41 | 8.8E−09 |
| cg27384074 | OPA3 | 0.55 | 1.7E−07 |

CpGs reaching significance of $p < 1 \times 10^{-6}$ were shown in the table.

given our study is observational. Comparing our study with evidence from MR studies which are less likely to be confounded (*Reed et al., 2017*; *Gill et al., 2018*; *Bell et al., 2018*; *Chen et al., 2019*), we found a consistent direction of association for dairy intake (*Huang et al., 2018b*) and binge eating (*Reed et al., 2017*) being associated with higher BMI. Evidence for some exposures, such as tea and chocolate consumption, is still lacking; further MR studies are needed to assess causality.

## Implications

In this study, we not only confirmed established risk factors, such as maternal second-hand smoking, but also identified several factors not reported or not examined in previous EWAS in western countries (*Vrijheid et al., 2020*; *Uche et al., 2020*), such as ASB consumption and soymilk intake. The comparison with RCTs or MR studies to some extent supports a role of dairy intake and binge eating, suggesting these factors or their drivers (e.g., sex hormones as a potential driver of binge eating) might be considered as potential targets for intervention. Other factors, such as soymilk intake, need to be tested in RCTs. We also identified several methylation loci related to obesity. Our study based on the unique setting in Hong Kong, provides potential drivers of obesity applicable to Hong Kong Chinese, with relevance to health policy interventions and future research.

## Conclusions

This study takes advantage of the unique setting of Hong Kong and provides more insight about the role of environmental exposures and epigenetics in early life obesity. If these associations are found to be causal, they may provide novel intervention targets to improve population health.

## Reporting

The study conforms with the STROBE checklist, which was attached as a supplementary file.

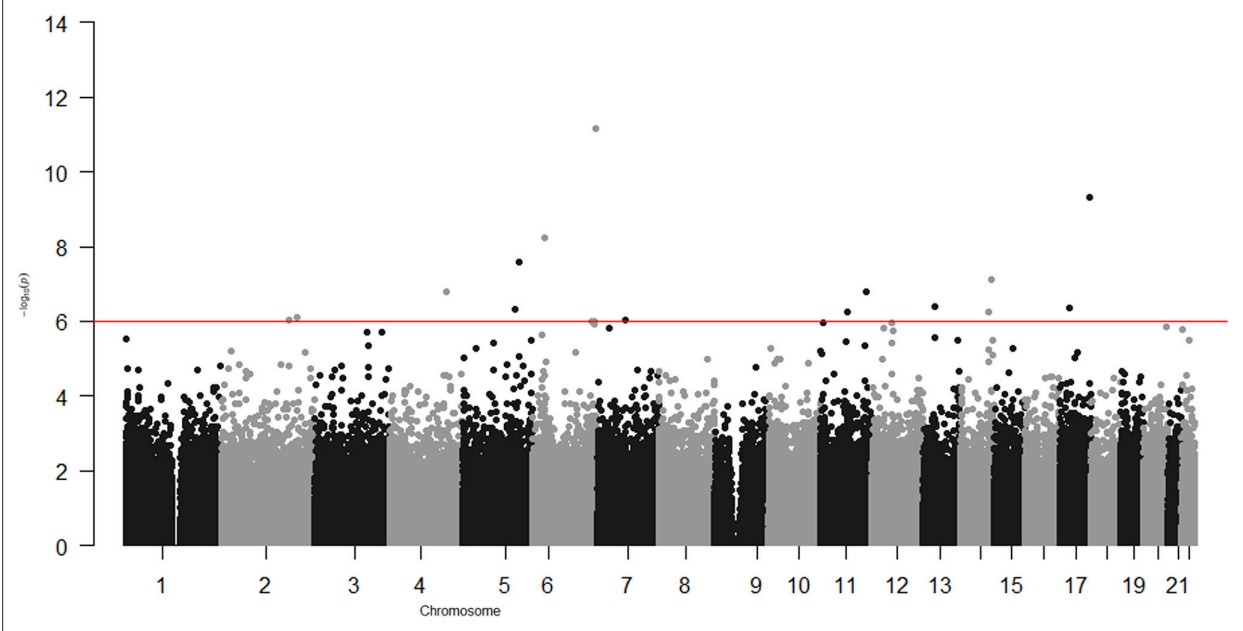

**Figure 3.** Epigenome-wide association with body mass index (BMI) at age ~23 years in 286 participants from Hong Kong's 'Children of 1997' birth cohort in the follow-up survey.

The online version of this article includes the following source data for figure 3:

**Source data 1.** The associations for depicting Figure 3 in the study.

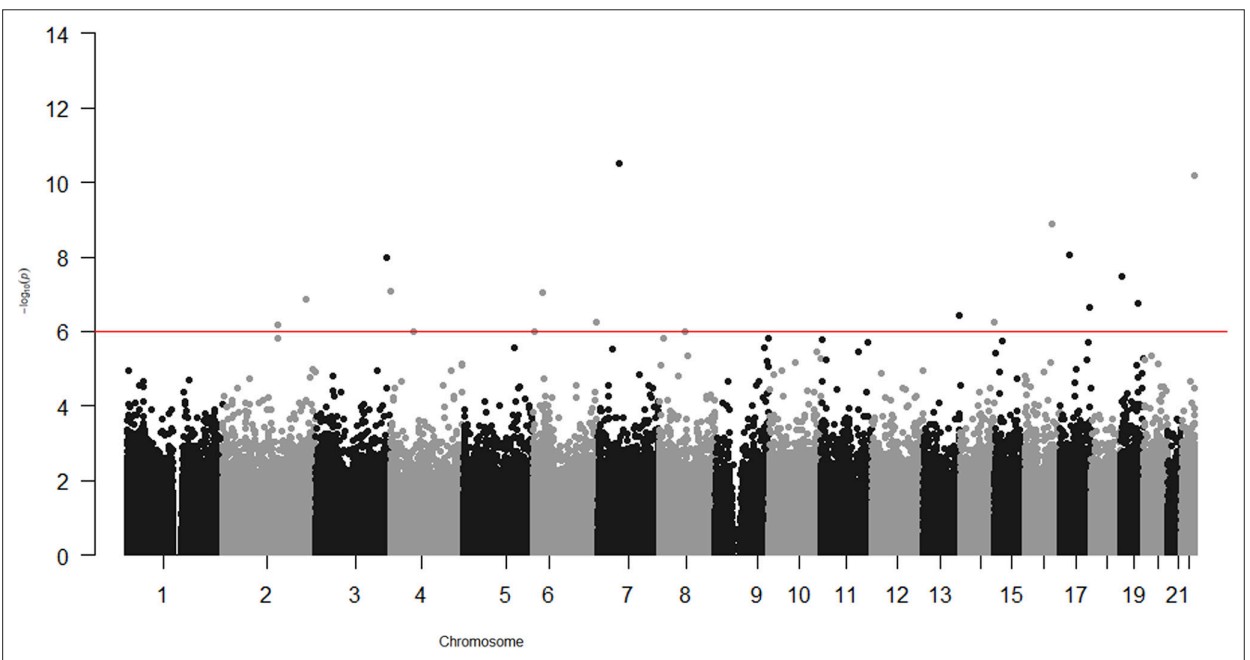

**Figure 4.** Epigenome-wide association with waist–hip ratio (WHR) at age ~23 years in 286 participants from Hong Kong's 'Children of 1997' birth cohort in the follow-up survey. Source code file: source code for the environment- and epigenome-wide association study analyses.

The online version of this article includes the following source data for figure 4:

**Source data 1.** The associations for depicting Figure 4 in the study.

## Acknowledgements

This study was supported by the Health and Medical Research Fund Research Fellowship, Food and Health Bureau, Hong Kong SAR Government (#04180097). The DNA extraction was supported by CFS-HKU1. We would like to sincerely thank Food and Health Bureau for funding this project. We would also like thank all the participants and research staff in this project.

## Additional information

### Competing interests

Benjamin John Cowling: Benjamin Cowling received research funding from Fosun Pharma, and received honoraria from AstraZeneca, Fosun Pharma, GlaxoSmithKline, Moderna, Pfizer, Roche and Sanofi Pasteur. The author has no other competing interests to declare. The other authors declare that no competing interests exist.

### Funding

| Funder | Grant reference number | Author |
| --- | --- | --- |
| Health and Medical Research Fund | 04180097 | Jie Zhao |
| Health and Medical Research Fund | CFS-HKU1 | Shiu Lun Ryan Au Yeung |

The funders had no role in study design, data collection, and interpretation, or the decision to submit the work for publication.

### Author contributions

Jie Zhao, Conceptualization, Resources, Data curation, Supervision, Investigation, Methodology, Writing – original draft, Writing – review and editing; Bohan Fan, Formal analysis, Visualization, Writing – original draft; Jian Huang, Formal analysis, Validation; Benjamin John Cowling, Shiu Lun Ryan Au Yeung, Andrea Baccarelli, Investigation, Writing – review and editing; Gabriel M Leung, Resources, Methodology, Writing – review and editing; C Mary Schooling, Resources, Investigation, Writing – review and editing

### Author ORCIDs

Jie Zhao  http://orcid.org/0000-0002-1564-0057
Bohan Fan  http://orcid.org/0000-0002-5349-9461
Benjamin John Cowling  http://orcid.org/0000-0002-6297-7154
Shiu Lun Ryan Au Yeung  http://orcid.org/0000-0001-6136-1836
C Mary Schooling  http://orcid.org/0000-0001-9933-5887

### Ethics

This study complies with the Declaration of Helsinki. Since our participants are children, informed (non-written) consent for the original survey and subsequent record linkage was obtained from the parents, next of kin, caretakers, or guardians (informants) on behalf of the participants by the informant agreeing and subsequently completing the questionnaire at enrollment, this manner of obtaining consent was approved by The University of Hong Kong Medical Faculty Ethics Committee over 20 years ago. Informed written consent for subsequent surveys and in-person follow-up was obtained from a parent or guardian, or at ages 18+ years from the participant. Ethical approval for this study, including the follow-up survey at ~23 years and comprehensive health-related analyses, was obtained from the University of Hong Kong-Hospital Authority Hong Kong West Cluster, Joint Institutional Review Board, Hong Kong Special Administrative Region, China (reference numbers: UW13-367; UW19-367).

### Decision letter and Author response

Decision letter https://doi.org/10.7554/eLife.82377.sa1
Author response https://doi.org/10.7554/eLife.82377.sa2

## Additional files

### Supplementary files

- MDAR checklist
- Supplementary file 1. Categorization and data sources for exposures.
- Supplementary file 2. Associations of selected exposures with BMI after adjusting for time difference in participants of Hong Kong's "Children of 1997" birth cohort.
- Source code 1. Main analysis code.

### Data availability

The data used in this study are based on the 'Children of 1997' Birth cohort, maintained by the School of Public Health, The University of Hong Kong. With the approved ethics for this study, the individual participant data cannot be made freely available online. Interested parties can access the data used in this study upon reasonable request, with approval by the birth cohort team. As part of this process, researchers will be required to submit a project proposal for approval, to ensure the data is being used responsibly, ethically, and for scientifically sound projects. Requesters should be employees of a recognized academic institution, health service organization, or charitable research organization with experience in medical research. Requestors should be able to demonstrate, through their peer-reviewed publications in the area of interest, their ability to carry out the proposed study. Source data files have been uploaded for each of the results figures (Figures 1–4) showing the model summary data for plotting the Manhattan plots in environment-wide and epigenome-wide associations with BMI and WHR. Source code for the analyses has been uploaded as Source Code.

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
