## [Editor Report]

This study presents a valuable finding on the association of environmental and epigenomic factors with obesity in the adolescent population, using a specialized cohort. The evidence supporting the claims of the authors is solid, although this study is inevitably open to residual confounding because of the limitation of observational study. The work will be of interest to both clinicians and researchers working on obesity and related metabolic disorders.

---

## [Decision Letter]

**Decision letter after peer review:**

Thank you for submitting your article "Environment-wide and epigenome-wide association study of adiposity in "Children of 1997" birth cohort" for consideration by *eLife*. Your article has been reviewed by 2 peer reviewers, and the evaluation has been overseen by a Reviewing Editor and Mone Zaidi as the Senior Editor. The reviewers have opted to remain anonymous.

Essential revisions:

1. The authors analyzed the association between exposures and obesity using linear regression. It is recommended to additionally perform logistic regression by converting BMI and waist circumference into binary dependent variables. Appropriate scientific evidence for cut-off values of BMI and waist circumference in adolescence and adulthood, respectively, should be provided. Regarding waist circumference, it is necessary to consider the cut-off value of each male and female.

2. The authors used a false discovery rate (FDR)<0.05 as the significance threshold for the epigenome-wide association study. However, considering the previous studies including 'Rakyan VK, et al. Epigenome-wide association studies for common human diseases. Nat Rev Genet 2011', it is appropriate to use a significance threshold between E-08 to E-06 for epigenome-wide association studies. If the sample size is not sufficient to proceed with the analysis using these significance thresholds, it is appropriate to delete the epigenome-wide association study from the manuscript.

3. In the limitations of this study, the authors used the following expression: "… despite the minimal level of confounding in this cohort…"

In presenting the limitations of this study, the authors used the following expression: "despite the minimal level of confounding in this cohort"

Considering the following points, this expression is wrong:

This is an observational study, and a large amount of confounding effect should be considered on the association between obesity and exposures analyzed in this study including diet, physical activity, mood and feelings, family history, and socioeconomic status. Since the confounders considered in the multivariable analysis are very limited, the possibility of residual confounding remains large. These limitations should be detailed in the discussion.

4. Since the outcome variables were measured by body mass index and waist-hip ratio, it is more appropriate to use the term "obesity" rather than "adiposity". It is recommended that the term "adiposity" used in the manuscript be changed to "obesity".

5. Only part of the analysis code is presented, not the entire code. The whole analysis code used for the actual analysis needs to be provided.

6. Provide information regarding definition of obesity according to age.

7. Provide general population characteristics of each age group (height, weight and z-score percentiles)

8. Provide the time difference between the age at the time of physical measurement and age at the exposure (survey period) and consider to include as variable.

9. Provide the basis for expressing onset and at end of puberty.

10. Change the term 'health' to another term as it is inappropriate to list (diabetes, growth problem…) etc.

11. Modify the way the table is organized and correct the order of variable description in table. For example, make two tables 4 and 5 into one, and check the order of enumeration of variables (the order of variables is different while comparing similar contents by age group, it must be corrected because it reduces readability)

*Reviewer #1 (Recommendations for the authors):*

1. The authors analyzed the association between exposures and obesity using linear regression. It is recommended to additionally perform logistic regression by converting BMI and waist circumference into binary dependent variables. Appropriate scientific evidence for cut-off values of BMI and waist circumference in adolescence and adulthood, respectively, should be provided. Regarding waist circumference, it is necessary to consider the cut-off value of each male and female.

2. The authors used a false discovery rate (FDR)<0.05 as the significance threshold for the epigenome-wide association study. However, considering the previous studies including 'Rakyan VK, et al. Epigenome-wide association studies for common human diseases. Nat Rev Genet 2011', it is appropriate to use a significance threshold between E-08 to E-06 for epigenome-wide association studies. If the sample size is not sufficient to proceed with the analysis using these significance thresholds, it is appropriate to delete the epigenome-wide association study from the manuscript.

3. In the limitations of this study, the authors used the following expression: "… despite the minimal level of confounding in this cohort…"

In presenting the limitations of this study, the authors used the following expression: "despite the minimal level of confounding in this cohort"

Considering the following points, this expression is wrong:

This is an observational study, and a large amount of confounding effect should be considered on the association between obesity and exposures analyzed in this study including diet, physical activity, mood and feelings, family history, and socioeconomic status. Since the confounders considered in the multivariable analysis are very limited, the possibility of residual confounding remains large. These limitations should be detailed in the discussion.

4. Since the outcome variables were measured by body mass index and waist-hip ratio, it is more appropriate to use the term "obesity" rather than "adiposity". It is recommended that the term "adiposity" used in the manuscript be changed to "obesity".

5. Only part of the analysis code is presented, not the entire code. The whole analysis code used for the actual analysis needs to be provided.

*Reviewer #2 (Recommendations for the authors):*

Please provide information regarding definition of adiposity according to age.

Please provide general population characteristics of each age group (height, weight and z-score percentiles)

Please provide the time difference between the age at the time of physical measurement and age at the exposure (survey period) and consider to include as variable.

Please provide the basis for expressing onset and at end of puberty.

Please changed the term 'health' to another term as it is inappropriate to list (diabetes, growth problem…) etc.

Please modify the way the table is organized and correct the order of variable description in table. For example, make two tables 4 and 5 into one, and check the order of enumeration of variables (the order of variables is different while comparing similar contents by age group, it must be corrected because it reduces readability)

---

## [Author Response]

Reviewer #1 (Recommendations for the authors):1. The authors analyzed the association between exposures and obesity using linear regression. It is recommended to additionally perform logistic regression by converting BMI and waist circumference into binary dependent variables. Appropriate scientific evidence for cut-off values of BMI and waist circumference in adolescence and adulthood, respectively, should be provided. Regarding waist circumference, it is necessary to consider the cut-off value of each male and female.

Thank you very much for your helpful comments. In the environment-wide association study, we used BMI, a continuous outcome, to maximize power in the selection of exposures passing Bonferroni-corrected significance. Then we examined the associations of these exposures with BMI after controlling for confounders in multivariable regression. Following your suggestions, in the multivariable regression, we converted BMI into a binary outcome (i.e., obesity) and performed logistic regression adjusted for potential confounders. We adopted the obesity cut-off values of BMI for different age groups with sex-specific cut-offs (1):

– BMI ≥ 20.89 kg/m2 for boys and BMI ≥ 21.20 kg/m2 for girls at age 11.5

– BMI ≥ 24.73 kg/m2 for men and BMI ≥ 24.85 kg/m2 for women at age 17.6

The directions of associations are consistent with those using BMI as continuous outcome, the comparisons are shown in Table 2 and Table 4.

It would be interesting to explore the associations with central obesity as a binary outcome as well, however, given the lack of high-quality evidence about the cut-off values for waist circumference and waist-to-hip ratio in Asian children and adolescents, we did not perform logistic regression on central obesity risk. We have revised the methods and results:

(Methods, paragraph 2)

We added “Given BMI has a more accepted cut-off value than WHR for children, we classified obesity as BMI ≥ 20.89 kg/m2 for boys and BMI ≥ 21.20 kg/m2 for girls at age 11.5; and BMI ≥ 24.73 kg/m2 for men and BMI ≥ 24.85 kg/m2 for women at age 17.6 (1).”

We updated Tables 2 and Table 4:

(Results, paragraph 2)

“Except for paternal diabetes which was not significant, the rest of exposures all showed consistent associations with obesity risk (Table 2).”

(Results, paragraph 3)

“Regarding the association with obesity risk at 17.6 years, birth weight, being a twin, maternal second-hand smoking, physical activity and energy drinks intake were not significant but the directions of associations are consistent (Table 4).”

We added in the discussion:

(Discussion, Strengths and Limitations)

“Third, we obtained similar results for BMI and obesity risk, however, given the lack of high-quality evidence about the cut-off values for waist circumference and waist-to-hip ratio in Asian children and adolescents, we did not perform logistic regression on central obesity risk.”

2. The authors used a false discovery rate (FDR)<0.05 as the significance threshold for the epigenome-wide association study. However, considering the previous studies including 'Rakyan VK, et al. Epigenome-wide association studies for common human diseases. Nat Rev Genet 2011', it is appropriate to use a significance threshold between E-08 to E-06 for epigenome-wide association studies. If the sample size is not sufficient to proceed with the analysis using these significance thresholds, it is appropriate to delete the epigenome-wide association study from the manuscript.

Thank you for your helpful suggestion. We have changed the threshold to 1×10-6 in the epigenome-wide association study, and updated the methods and results accordingly.

(Methods-statistical analysis, paragraph 2)

We revised to “The significance was considered as p<1×10-6, genome-wide significance (5×10-8) was not used given the relatively small sample size in the epigenome-wide association study.”

(Results, paragraph 5)

We revised to “In the epigenome-wide association study of 286 participants we identified 17 CpGs for BMI at ~23 years in the genes RBM16, SCN2B, SLC24A4, TECPR2, KSR1, RPTOR, GTF3C3, ZNF827, TXNDC15, C2, and RPS6KA2 and 17 for WHR in the genes LANCL2, C6orf195, MIR4535, CTRL, LYRM9, DCDC2, DIRC3, RPS6KA2, LPP, NFIC, MIR7641-2, ZNF141, RNF213, and OPA3 (Tables 7-8 and Figures 3-4).”

3. In the limitations of this study, the authors used the following expression: "… despite the minimal level of confounding in this cohort…"In presenting the limitations of this study, the authors used the following expression: "despite the minimal level of confounding in this cohort"Considering the following points, this expression is wrong:This is an observational study, and a large amount of confounding effect should be considered on the association between obesity and exposures analyzed in this study including diet, physical activity, mood and feelings, family history, and socioeconomic status. Since the confounders considered in the multivariable analysis are very limited, the possibility of residual confounding remains large. These limitations should be detailed in the discussion.

Thank you very much for your very helpful suggestion, we totally agree that our study is an observational study and inevitably open to residual confounding, although we have controlled for sex, housing type at birth, household income at birth, maternal second-hand smoking during pregnancy, maternal age at birth, maternal education, maternal birthplace, and the interaction of maternal education with maternal birthplace. So, we also compared the associations from multivariable regression with available evidence from Mendelian randomization studies, which uses genetic variants as instrument to minimize confounding. We have added more in the discussion.

(Discussion, Strengths and Limitations)

We revised from: “Finally, despite the minimal level of confounding in this cohort for some key exposures, such as breastfeeding, residual confounding likely exists, so the associations are not definitive.”

To: “Finally, although we controlled for several confounders in the multivariable analysis, residual confounding may still exist given our study is observational. Comparing our study with evidence from MR studies which are less likely to be confounded (2-5), we found a consistent direction of association for dairy intake (39) and binge eating (40) being associated with higher BMI. Evidence for some exposures, such as tea and chocolate consumption, is still lacking; further MR studies are needed to assess causality.”

4. Since the outcome variables were measured by body mass index and waist-hip ratio, it is more appropriate to use the term "obesity" rather than "adiposity". It is recommended that the term "adiposity" used in the manuscript be changed to "obesity".

Thank you very much for your very helpful suggestion. We have changed the term "adiposity" to "obesity" in the manuscript throughout.

5. Only part of the analysis code is presented, not the entire code. The whole analysis code used for the actual analysis needs to be provided.

Thank you very much for your comments. We previously showed part of the analysis code in the environment-wide association study because the analysis code is similar for BMI and WHR, and for different ages. As you suggested, we have uploaded the whole analysis code along with the revised manuscript.

Reviewer #2 (Recommendations for the authors):Please provide information regarding definition of adiposity according to age.

Thank you very much for your helpful suggestion. We added the definition of obesity by the World Health Organization (WHO), and also addressed the cut-off values specific to Asian population by age and by sex.

(Introduction, paragraph 1)

“Obesity is a well-established risk factor for multiple chronic diseases, including cardiovascular disease, diabetes, and cancer (6). According to the World Health Organization (WHO), obesity is defined as “abnormal or excessive fat accumulation that presents a risk to health” (7). Body mass index (BMI) is the most appropriate measure for overweight and obesity because the cut-offs account for age, sex, and ethnicity (7). Obesity has increased substantially in many settings, including Hong Kong…”

We clarified the obesity cut-off by adding the following sentence:

(Methods, Outcomes)

“Given BMI has a more accepted cut-off value than WHR for children, we classified obesity as BMI ≥ 20.89 kg/m^2^ for boys and BMI ≥ 21.20 kg/m^2^ for girls at age 11.5; and BMI ≥ 24.73 kg/m^2^ for men and BMI ≥ 24.85 kg/m^2^ for women at age 17.6 (1).”

Please provide general population characteristics of each age group (height, weight and z-score percentiles)

Thank you very much for your helpful suggestion. We have added information of each age group (height, weight, etc) in Table 1.

Please provide the time difference between the age at the time of physical measurement and age at the exposure (survey period) and consider to include as variable.

Thank you very much for your suggestions. We understand that the time difference between the age at the time of physical measurement and age at the exposure (time of survey period) is an important aspect and sometimes may lead to imprecision in measurement in large cohort studies. The Biobank Clinical follow-up at 17.6 years involved collecting questionnaires from children and parents, and anthropometric measurements from participants in one visit, so the time difference is minimal. However, for exposures collected at baseline and the Survey I, there may be a difference between the age at which physical measurements were taken and age at the exposure. To address this issue, we included a variable for the time differences associated with these exposures, and results were similar after including the time difference variable (Supplementary file 2). We added the following sentences in the Methods and Results:

(Methods, Statistical analysis)

“Additionally, to account for the time lag between the age at which physical measurements were taken and age at exposure collection, we included a time difference variable in the adjusted models for BMI.”

(Results, paragraph 3)

“The associations of selected exposures with BMI at ~11.5 years and at 17.6 years were similar after adjusting for the time difference between age at anthropometric measurements and age of exposure collection (Supplementary file 2).”

(Discussion, Strength and Limitations)

We added: “After accounting for the time difference between age at anthropometric measurements and age of exposure collection, we also obtained similar results.”

Please provide the basis for expressing onset and at end of puberty.

Thank you very much for raising this question. We considered puberty because puberty is an important stage which involves a re-orientation from childhood priorities to adulthood (24). Exposures associated with adiposity at puberty may be important for health in later life (25). We considered onset and at the end of puberty separately, because due to the changes in hormones during puberty, exposures related to obesity at the onset and at the end of puberty may be different, the same exposure may also have different effects on obesity. We added more explanation in the Introduction section:

(Introduction, paragraph 4)

“We focused on puberty because it is an important stage involving a re-orientation from childhood priorities to adulthood (24). Exposures associated with obesity at puberty may be important for health in later life (25). Considering that exposures related to obesity at the outset and at the end of puberty may be different, and the associations of the same exposure with obesity may vary by age, we conducted the EWAS at different ages.”

Please changed the term 'health' to another term as it is inappropriate to list (diabetes, growth problem…) etc.

Thank you very much for pointing out the inappropriate term, we have incorporated your suggestion throughout the manuscript by changing ‘health’ into ‘health status’, with a note that the term here refers to physical health condition. Please refer to the tracked changes in the revised manuscript (Exposures and categorization of Methods) and revised Supplementary file 1, Figures 1 and 2.

(Methods, Exposures and categorization)

We revised to “The exposures considered … child’s health status (referring to physical health condition; detail questionnaire questions about this can be found in Supplementary file 1), parents’ health status,…”

Please modify the way the table is organized and correct the order of variable description in table. For example, make two tables 4 and 5 into one, and check the order of enumeration of variables (the order of variables is different while comparing similar contents by age group, it must be corrected because it reduces readability)

Thank you very much for this suggestion, we have incorporated your suggestion and reorganized the tables by combining Tables 2 and 3, Tables 4 and 5, Tables 6 and 7, Tables 8 and 9.

References

1. Cole TJ. Establishing a standard definition for child overweight and obesity worldwide: international survey. BMJ. 2000;320(7244):1240-.

2. Reed ZE, Micali N, Bulik CM, Davey Smith G, Wade KH. Assessing the causal role of adiposity on disordered eating in childhood, adolescence, and adulthood: a Mendelian randomization analysis. Am J Clin Nutr. 2017:ajcn154104.

3. Gill D, Brewer CF, Del Greco MF, Sivakumaran P, Bowden J, Sheehan NA, et al. Age at menarche and adult body mass index: a Mendelian randomization study. Int J Obes. 2018;42(9):1574-81.

4. Bell JA, Carslake D, Wade KH, Richmond RC, Langdon RJ, Vincent EE, et al. Influence of puberty timing on adiposity and cardiometabolic traits: A Mendelian randomisation study. PLoS Med. 2018;15(8):e1002641.

5. Chen YC, Fan HY, Yang C, Hsieh RH, Pan WH, Lee YL. Assessing causality between childhood adiposity and early puberty: A bidirectional Mendelian randomization and longitudinal study. Metabolism. 2019;100:153961.

6. Gallagher EJ, LeRoith D. Obesity and Diabetes: The Increased Risk of Cancer and Cancer-Related Mortality. Physiol Rev. 2015;95(3):727-48.

7. World Health Organization. Regional Office for the Western P. The Asia-Pacific perspective : redefining obesity and its treatment: Sydney : Health Communications Australia; 2000 2000.